# Elevated miR-29c-5p Expression in Nipple Aspirate Fluid Is Associated with Extremely High Mammographic Breast Density

**DOI:** 10.3390/cancers14153805

**Published:** 2022-08-05

**Authors:** Tessa A. C. M. Vissers, Leonie Piek, Susana I. S. Patuleia, Aafke J. Duinmeijer, Marije F. Bakker, Elsken van der Wall, Paul J. van Diest, Carla H. van Gils, Cathy B. Moelans

**Affiliations:** 1Department of Pathology, University Medical Center Utrecht, 3508 GA Utrecht, The Netherlands; 2Department of Medical Oncology, University Medical Center Utrecht, 3508 GA Utrecht, The Netherlands; 3Department of Epidemiology of the Julius Center for Health Sciences and Primary Care, University Medical Center Utrecht, 3508 GA Utrecht, The Netherlands

**Keywords:** miRNA, mammographic density, breast cancer, nipple aspirate fluid

## Abstract

**Simple Summary:**

High mammographic density is a known risk factor for breast cancer. However, the underlying mechanisms of high mammographic density development and breast cancer are unknown. MicroRNAs are potential biomarkers indicative of carcinogenesis and can be assessed in nipple aspirate fluid. We used nipple aspirate fluid from women with very low and extremely high mammographic density to examine differences in expression of multiple miRNAs between both extremes in the spectrum of mammographic density. We found that hsa-miR-29c-5p was upregulated in an extremely high mammographic density context and potential targets were identified that might provide clues of the relationship between high mammographic density and breast cancer risk. Understanding the relationship between high mammographic density and breast cancer is of great value for early breast cancer diagnosis and treatment. With our research we provide new insight into this relationship and further research could determine the effects of dysregulated hsa-miR-29c-5p on the identified candidate targets.

**Abstract:**

High mammographic density (MD) is associated with an increased risk of breast cancer, however the underlying mechanisms are largely unknown. This research aimed to identify microRNAs (miRNAs) that play a role in the development of extremely dense breast tissue. In the discovery phase, 754 human mature miRNAs were profiled in 21 extremely high MD- and 20 very low MD-derived nipple aspirate fluid (NAF) samples from healthy women. In the validation phase, candidate miRNAs were assessed in a cohort of 89 extremely high MD and 81 very low MD NAF samples from healthy women. Independent predictors of either extremely high MD or miRNA expression were identified by logistic regression and linear regression analysis, respectively. mRNA targets and pathways were identified through miRTarBase, TargetScan, and PANTHER pathway analysis. Statistical analysis identified four differentially expressed miRNAs during the discovery phase. During the validation, linear regression (*p* = 0.029; fold change = 2.10) and logistic regression (*p* = 0.048; odds ratio = 1.38) showed that hsa-miR-29c-5p was upregulated in extremely high MD-derived NAF. Identified candidate mRNA targets of hsa-miR-29c-5p are CFLAR, DNMT3A, and PTEN. Further validation and exploration of targets and downstream pathways of has-miR-29c-5p will provide better insight into the processes involved in the development of high MD and in the associated increased risk of breast cancer.

## 1. Introduction

Mammography is currently the primary screening method for timely detection and treatment of breast cancer. The extent of fibro-glandular versus fatty tissue on a mammogram can classify breast tissue into four categories of density, from A to D. A higher mammographic density (MD) reflects a high quantity of stromal and glandular tissue and a low quantity of adipose tissue as opposed to lower density breasts, which are almost entirely made up of fatty breast tissue [1]. Low MD (category B) and high MD (category C) are the most common, both occurring in about 40% of women, while very low (category A) and extremely high MD (category D) both occur in approximately 10% of women [2,3]. Enhanced MD is therefore present in 50% of all women and not only reduces mammographic sensitivity by masking tumours, but also strongly and independently predicts breast cancer risk [3]. Extremely high MD is associated with a 4–6-fold higher risk of developing breast cancer compared to very low MD, and 42.8% of premenopausal breast cancer cases are believed to be attributable to having dense breasts (categories C and D) [3,4,5,6]. Although multiple factors including extracellular matrix (ECM) proteins [3], small leucine rich proteoglycans [7], glycoproteins such as CD36 [8], fibroblasts [9] and immune cells [10] seem to be altered in high MD, the underlying mechanism of high breast density is still far from understood.

MicroRNAs (miRNAs) might be among the players involved in the regulation of (extra)cellular processes that modulate MD, such as fat accumulation and connective tissue density, and have an established role in carcinogenesis [11]. miRNAs are small, non-coding RNAs (~18–24 nucleotides long) that regulate gene expression at the post-transcriptional level, usually by binding to the three prime untranslated region (3′UTR) of messenger RNAs (mRNA) [11]. This interaction destabilizes the mRNA or suppresses protein translation and therefore affects gene expression. miRNAs represent potentially applicable biomarkers due to their structural stability in biological fluids and their ability to be extracted and measured from limited sample volumes [12].

In the context of MD, the assessment of miRNAs in nipple aspirate fluid (NAF) may be particularly relevant [13]. NAF contains breast-specific miRNAs, actively or passively secreted from the breast microenvironment into the breast ducts. Acquisition of this fluid is feasible by means of non-invasive oxytocin-supported vacuum-aspiration [14]. As part of the nationwide Dense tissue and Early breast Neoplasm Screening (DENSE) and DENSE-on trials, NAF was acquired from a subset of healthy women undergoing population screening with extremely high and very low MD breasts, respectively [15,16]. Here, we aimed to identify NAF-derived miRNAs that are aberrantly expressed between these two extremes within the breast density spectrum. In addition, we aimed to elucidate relevant cellular pathways affected by the dysregulated miRNAs. Knowledge of the key players that establish high MD is needed to understand the etiological processes that link breast density to breast cancer. Eventually, miRNAs might enable risk prediction of breast cancer in women with high MD and they, or their targets, could be exploited for the development of targeted therapies to modify this density-triggered risk of developing breast cancer.

## 2. Materials and Methods

### 2.1. Nipple Aspirate Fluid Collection and Processing

A total of 110 healthy women with extremely high MD breasts (classified as an ACR4 or ‘D’ on a mammogram using Volpara software (Volpara Solutions)) and 101 healthy women with very low MD breasts (ACR1 or ‘A’), representing the two extreme categories within the four-category MD spectrum (A to D [1]), were included in this study after informed consent was obtained. Sample size was determined by available samples. These women were participants of the larger, nationwide DENSE trial (NCT01315015; [15,16]) and the DENSE-on low biobanking study, approved by the Institutional Review Boards within the participating hospitals, and the UMC Utrecht Biobank Research Ethics Committee (TCBio; biobank study number 14–467 approved on 17 June 2015). All women were aged between 50–74 years and had a negative screening mammography and/or an MRI without abnormalities at the time of NAF acquisition. Median age was 55 years old, with an interquartile range (IQR) of 52–58.5. In the group of women with entirely fatty breasts, age was restricted to 50–60 years to prevent the effect of age on mammographic density. Information on lifestyle, reproductive characteristics and medical history were obtained by means of a questionnaire.

Bilateral NAF samples were acquired between June 2015 and March 2016 at several Dutch hospitals or at the home of the study subjects by trained research nurses. Collection was performed using a vacuum device after nasal oxytocin administration, as described earlier by Suijkerbuijk et al. [14]. The collected fluid was conserved in a buffer solution (50 mM Tris pH 8.0, 150 mM NaCl, 2 mM EDTA) at −80 °C until required for analysis. Details concerning sample collection success, sample volume (between 5–100 µL) and sample color were registered. Analyses were conducted in two main phases, a discovery phase (21 extremely high and 20 very low MD NAF samples) and a validation phase (89 extremely high and 81 very low MD NAF samples). The Biospecimen Reporting for Improved Study Quality (BRISQ) guidelines were taken into account [17].

### 2.2. RNA Isolation, Reverse Transcription, and Pre-Amplification

Total RNA was extracted from pooled NAF samples (intra-individual samples from left and right breast combined, for discovery) or unilateral NAF samples (left or right breast at random, for validation) according to the manufacturer’s protocol using the AllPrep DNA/RNA/miRNA Universal Kit (Qiagen, Hilden, Germany). All isolations were performed starting with 20 µL of pooled NAF or 10 µL of unilateral NAF if available. Pooled NAF was used in the discovery phase to ensure sufficient RNA for profiling. Non-human synthetic ath-miR159a (with a 5′ phosphate) was spiked in as procedural control at 300 pg by pre-mixing with RLT plus lysis buffer. Total RNA was eluted in 30 µL RNAse-free water. RNA concentrations were determined with the Qubit RNA HS Assay Kit (Invitrogen, Q32852, Waltham, MA, USA) measured by Qubit 3.0 (ThermoFisher Scientific, Waltham, MA, USA) fluorometric quantification.

A uniform RNA sample concentration was obtained by dilution in RNAse-free water to 4 ng/µL or to 2.5 ng/µL for the discovery phase and the validation phase, respectively. First, according to the manufacturer’s instructions, 8 ng (discovery phase) or 5 ng (validation phase) of total RNA was poly-A tailed. After adaptor ligation and reverse transcription, cDNA was pre-amplified for nineteen cycles using the TaqMan Advanced miRNA cDNA Synthesis Kit (ThermoFisher Scientific, Waltham, MA, USA) on a Veriti 96-well thermal cycler (ThermoFisher Scientific, Waltham, MA, USA). The pre-amplification product was subsequently diluted 20× in 0.1× Tris buffer, pH 8.0 and stored at −20 °C until qPCR.

### 2.3. Discovery Phase: Taqman OpenArray Profiling Analysis of Nipple Aspirate Fluid

The expression levels of 754 human mature miRNAs (Appendix A), that were functionally validated with miRNA artificial templates, were profiled using the fixed-content TaqMan OpenArray Human Advanced MicroRNA Panel on a QuantStudio 12K Flex system (ThermoFisher Scientific, Waltham, MA, USA). The samples were loaded from a 384-well plate onto the TaqMan OpenArray Human Advanced MicroRNA Panel array slide using the OpenArray AccuFill system. Relative threshold values (CRT [18]) were automatically generated using the ThermoFisher Cloud system (https://apps.thermofisher.com/ (accessed on 24 November 2020)). These were proven to be more robust than baseline threshold values for analysing data generated using nanolitre fluidics based OpenArray plates. Analysis settings included the following restrictions: a minimum CRT of 10, a minimum AMPSCORE (low signal in linear phase) of 1, a minimum calculated confidence in the quantification cycle (CQCONF) (Cq) value of 0.6, a maximum CRT of 30 with inclusion of maximum CRT in calculations. Additionally, all miRNA amplification plots were visually inspected on curve shape and signal timing. The global mean was used for normalization [19,20].

A technical validation (i.e., quality control of NAF profiling data using the same samples) was performed using 14 randomly selected individual Taqman advanced miRNA assays (hsa-miR-324-5p, hsa-miR-29b-3p, hsa-miR-19a-3p, hsa-miR-186-5p, hsa-miR-361-5p, hsa-miR-425-5p, hsa-miR-187-3p, hsa-miR-29a-3p, hsa-miR-660-5p, hsa-miR-155-5p, hsa-miR-181a-5p, hsa-miR-222-3p, hsa-miR-339-5p, and hsa-miR-25-3p) (for assay IDs see Appendix A) and TaqMan Fast Advanced Master Mix (Thermofisher Scientific) according to the manufacturer’s instructions on a ViiA7 real-time PCR system (ThermoFisher Scientific).

Unsupervised hierarchical clustering of NAF samples based on their miRNA expression pattern was performed using the web tool ClustVis (http://biit.cs.ut.ee/clustvis (accessed on 12 December 2021)) [21]. Missing values were automatically imputed by ClustVis.

### 2.4. Validation Phase: Individual TaqMan Advanced miRNA qPCR Assays

Validation of differentially expressed candidate miRNAs identified in the discovery phase was performed in a larger, independent NAF cohort (*n* = 170, 89 extremely high and 81 very low MD samples). Individual TaqMan advanced mature miRNA assays for hsa-miR-92a-3p, hsa-miR-92b-3p, hsa-miR-22-5p, hsa-miR-29c-5p, hsa-miR-125a-5p (endogenous control identified by GeNorm analysis (https://genorm.cmgg.be/ (accessed on 24 November 2020)) [20]), and ath-miR159a (technical control) (for assay IDs see Appendix A) were used with TaqMan Fast Advanced Master Mix (ThermoFisher Scientific, Waltham, MA, USA) according to the manufacturer’s instructions on a ViiA7 real-time PCR system (ThermoFisher Scientific, Waltham, MA, USA). All qPCR reactions were performed in duplicate and interplate calibrator samples and non-template controls were used throughout. Cycle threshold (CT) values >35 were considered undetermined. The data were analyzed using comparative quantification [22]. Delta CT values (CT target miRNA minus CT endogenous control miRNA) were calculated and used for further statistical analysis with hsa-miR-125a-5p as endogenous control. Appendix A shows raw CT values per sample. Analyses were performed after replacement of missing miRNA values (missings in <3/4 miRNAs per sample) by the maximum CT value obtained for the particular miRNA +1 [23].

### 2.5. Statistical Analysis

Established risk factors for breast density and breast cancer were included in the statistical analysis. Body mass index (BMI) values, calculated based on self-reported height and weight, were derived from the moment of inclusion in the DENSE(-on) study. The baseline characteristics and reproductive information of the study subjects were analyzed as continuous data (for age, BMI, age at menarche and age at first live birth) or as dichotomous categorical data (for parity (nulliparous or parous) or having a first-degree family member with breast cancer (yes/no)).

Statistical analyses were performed using IBM SPSS Statistics for Windows version 27.0 (IBM Corp., Orchard Road Armonk, NY, USA) and RStudio version 1.3.1093 (Public-benefit corporation, Vienna, Austria). Normal distribution of the data was analyzed by visual inspection (histograms, residual, and Q-Q plots) and by means of a Kolmogorov–Smirnov test. Differences between patient characteristics were assessed by ANOVA and independent sample t-tests for normally distributed data, whereas Mann–Whitney U-tests were used for non-normally distributed data. For dichotomous data comparison, Pearson chi-square or Fisher’s exact tests were applied. A Pearson’s correlation test was performed to compare profiling and individual assay results within the discovery set, and to explore collinearity.

To identify differentially expressed miRNAs between extremely high and very low MD cohorts within the discovery set, a two-step approach was used. First, using the delta CT (DCT) values of 118 miRNAs expressed in >80% of discovery samples (>32/41), a linear regression was performed in SPSS with miRNA DCT value as outcome variable and BMI, dense category (extremely high or very low), parity (parous or nulliparous), age, and NAF color class as predictor variables (enter method, with variable entry at *p* < 0.05 and removal at *p* < 0.10). The latter (NAF color class) was included in the linear regression model because previous in-house data showed a significant association between these parameters and miRNA expression [24]. NAF color was classified into four categories (clear white/yellow, turbid white/yellow, bloody/orange/pink and green/brown), of which dummy variables were made. miRNAs with *p* < 0.20 for breast density as predictor were noted. Second, using the same 118 miRNAs, a Firth’s bias-reduced logistic regression was performed in RStudio (package logistf, version 1.24) using dense category as outcome, and the other variables (BMI, age, and NAF color) including miRNA DCT values as predictors. In this analysis, the NAF color variable was reduced to three categories: white/yellow, bloody/orange/pink and green/brown. Again, miRNAs with *p* < 0.20 for predicting dense category and their coefficients were noted. All miRNAs overlapping between both analyses were selected for subsequent validation experiments. To identify differentially expressed miRNAs between extremely high and very low MD cohorts within the validation set, we used the same approach. Fold change in extremely high versus very low MD and odds ratios with 95% confidence intervals (CI) were calculated for, respectively, the linear regression and the Firth’s bias-reduced logistic regression approach [22]. Here, miRNAs with *p* < 0.05 were considered significant. GraphPad Prism 8.3 for Windows (San Diego, CA, USA) was used for graphical visualization of the results.

### 2.6. Target Analysis of Differentially Expressed miRNAs

Established target analysis of validated differentially expressed miRNAs was performed in miRtarbase 9.0 (2021, Shenzhen, China) (https://mirtarbase.cuhk.edu.cn/ (accessed on 17 February 2022)) [25]. The search query included the miRNA names (hsa-miR-x-x) of the respective miRNA(s), and the data were selected based on strong experimental evidence which was defined as data derived from luciferase reporter assays, Western blot experiments or quantitative PCRs. Relevant predicted targets based on miRNA sequence were identified through TargetScan 8.0 (https://www.targetscan.org/ (accessed on 22 February 2022)) [26] and miRWalk (http://mirwalk.umm.uni-heidelberg.de/ (accessed on 22 February 2022)) [27]. Protein class, Gene Ontology (GO) biological process (BP) and Reactome pathways of established mRNA targets were explored via PANTHER 16.0 pathway analysis (http://pantherdb.org/ (accessed on 23 February 2022)) [28].

## 3. Results

### 3.1. Study Subject and NAF Characteristics Per Cohort

Baseline characteristics of extremely high and very low MD discovery and validation cohorts are shown in Table 1. Between the two extreme MD categories, significant differences were observed for the variables body mass index (BMI) (discovery and validation cohorts showed lower BMI in extremely high MD, both *p* < 0.0001), age at menarche (validation cohort older menarche age in extremely high MD cohort, *p* = 0.001) and NAF color (discovery cohort more bloody/orange/pink NAF samples in extremely high MD, *p* = 0.028).

No significant differences were observed between the discovery and validation cohorts concerning age at the time of NAF collection, BMI, age at first live birth, age at menarche, parity, and having a first-degree family member with breast cancer (Appendix A). NAF color was however significantly different (*p* < 0.0001) between discovery and validation cohorts. Total RNA concentrations in both cohorts varied between 2.5 and 134 ng/µL, with a median RNA concentration of 33 ng/µL in the discovery cohort and 5 ng/µL in the validation cohort (*p* < 0.0001).

### 3.2. Discovery: Four Differentially Expressed miRNAs between Extremely High and Very Low MD

Out of the 754 interrogated human mature miRNAs, 118 (16%) could reliably be detected in at least 80% (>32/41) of NAF samples. Four of these miRNAs significantly differed between extremely high and very low dense categories in linear regression as well as Firth’s corrected logistic regression, including the confounders BMI, age, parity, and NAF color class: hsa-miR-22-5p, hsa-miR-29c-5p, hsa-miR-92a-3p, and hsa-miR-92b-3p (Figure 1). Hsa-miR-22-5p and hsa-miR-29c-5p were upregulated in extremely high MD-derived NAF (FC 2.77 with *p* = 0.027 and FC 1.92 with *p* = 0.031, respectively) while hsa-miR-92a-3p and hsa-miR-92b-3p were downregulated in extremely high MD-derived NAF (FC 0.75 with *p* = 0.125 and FC 0.60 with *p* = 0.139, respectively). Firth’s corrected logistic regression supported the results of the linear regression, but was only able to include the confounders BMI, age, and NAF color class due to the small sample size (Appendix A). Hsa-miR-22-5p and hsa-miR-29c-5p were upregulated in extremely high MD-derived NAF (OR 1.90 (95% CI 0.80–6.31); *p* = 0.153 and OR 7.02 (95% CI 1.21–380.67); *p* = 0.026, respectively), whereas hsa-miR-92a-3p and hsa-miR-92b-3p were downregulated in extremely high MD-derived NAF (OR 0.15 (95% CI 0.01–1.18); *p* = 0.072 and OR 0.19 (95% CI 0.01–1.06); *p* = 0.059).

Unsupervised hierarchical clustering of NAF samples based on the expression level of the four candidate miRNAs (without correction for confounders) revealed two clusters, one cluster containing relatively more extremely high-density samples and one cluster containing relatively more very low-density samples (Appendix A).

The reliability of the microfluidics-based discovery was further technically validated with regular-volume qPCR, showing a fair to high correlation for all 14 randomly selected miRNAs (Pearson correlation coefficient range 0.37–0.93) (Appendix A).

### 3.3. Validation: Hsa-miR-29c-5p Is Differentially Expressed between Extremely High and Very Low MD

Eleven NAF samples (6.5% of the validation cohort) showed undetermined CT values for all four candidate miRNAs. Another three samples (1.8%) showed undetermined or late CT (>31) for at least three of the interrogated miRNAs, and these were excluded from further analysis. Statistical analysis was performed on the remaining 156 samples, and sporadic missings were replaced by the maximum CT value +1 per miRNA. Based on 146 samples (86% of total NAF validation cohort) in the complete regression model including BMI, age, parity and NAF color class, linear regression confirmed that the expression of one of the candidate miRNAs, hsa-miR-29c-5p, was significantly increased in an extremely high MD context (FC 2.1; *p* = 0.029) (Figure 2a). Firth’s corrected logistic regression supported the linear regression (OR = 1.38 (95% CI 1.00–1.98); *p* = 0.048) (Figure 2b). All four miRNAs were differentially expressed between NAF colors. Hsa-miR-29c-5p and hsa-miR-92b-3p were downregulated in parous versus nulliparous women (FC 0.46; *p* = 0.019, and FC 0.53; *p* = 0.016, respectively) while BMI and age had no influence on miRNA expression.

mRNA target analysis was performed to investigate the functional role of hsa-miR-29c-5p (Table 2). Hsa-miR-29c-5p has few established targets as most research so far has focused on the dominant miRNA arm, hsa-miR-29c-3p, and other family members such as miR-29a and miR-29b sharing the same seed region [29]. Established miR-29c-5p targets so far include DNMT3A [30], TMEM98 [31], and CPEB4 [32]. CFLAR, YY1, PTEN, and CD36 are potentially interesting targets in relation to MD based on computational methods. Considering the small number of established targets for this miRNA, no enrichment analysis could be performed. A graphical overview of the validated miRNA and its potential relation to high MD reflected by some of its established and predicted targets is provided in Figure 3.

## 4. Discussion

Mammographic breast density is one of the strongest, independent risk factors for breast cancer [1]. MiRNAs may be involved in the development of extremely high MD and can be studied in NAF as a representative liquid biopsy of the breast microenvironment. Here, we demonstrated elevated hsa-miR-29c-5p expression in two independent NAF cohorts acquired from women with extremely high MD, compared to very low MD.

Hsa-miR-29c-5p has been identified as one of the most significantly differentially expressed miRNAs between ER-positive and -negative breast tumors, demonstrating a strong partly GATA3-dependent upregulation in ER-positive/luminal subtype tumors compared to normal breast tissue and ER-negative tumors, and which is already observed in pre-invasive breast lesions [33,34]. The fact that miR-29c-5p dysregulation occurs early on during carcinogenesis is in line with its elevated expression in the context of extremely high MD. Other studies have, however, suggested a tumor suppressor rather than oncogenic role of miR-29-5p in (breast) cancer as this miRNA is generally more highly expressed in less aggressive/better prognosis subtypes, and overexpression in cancer cell lines typically inhibits metastasis, proliferation, and migration while increasing apoptosis [31,32,35]. Additionally, many of its established targets so far seem to contribute to a more cancerous phenotype [36,37,38,39,40]. Nevertheless, our finding of a higher NAF miR-29c-5p expression in the context of extremely high MD could be a consequence rather than a cause, aiming to suppress breast cancer in individuals with a higher breast cancer risk due to extremely high MD. Or, this miRNA could be selectively released into NAF leading to differing cellular and extracellular miRNA profiles. Interestingly, the miR-29 family has been reported to frequently reside in extracellular exosomes and upregulation of plasma exosomal miR-29c-5p has recently been suggested as a diagnostic biomarker for early lung carcinoma [41]. Furthermore, miR-29c-5p was upregulated in bladder serum cancer samples compared to normal samples and its dysregulation was correlated to advanced stage and poor outcome in bladder cancer patients [42].

To better understand the role of miR-29c-5p in MD, a target analysis was performed. Few robustly validated targets have been described for miR-29c-5p. Its expression in breast cancer was found negatively correlated with the mRNA and protein expression of the DNA methyltransferase DNMT3A, which can alter global DNA methylation levels and, amongst others, result in increased collagen type I (COL1A1) promoter activity, contributing to high MD [36]. An interesting predicted miR-29c-5p target based on computational algorithms, assuming an oncogenic role, is the apoptosis regulator protein CFLAR, which was previously identified as one of the plasma proteins negatively associated with area-based breast density [37]. CFLAR mRNA expression is also lower in breast cancer when compared to normal counterpart tissue [43]. MiR-29c-5p might also target PTEN. Low stromal PTEN expression is associated with multiple cancers, but also with high mammographic density through collagen deposition [44]. Murine PTEN knockout models demonstrate more collagen secretion, fiber formation and increased expression of the ECM regulator SPARC [44]. Furthermore, a mouse model with fibroblast-specific PTEN deletion resulted in an alteration of collagen organization, a promotion of collagen orientation in surrounding tumors, and an enhancement of highly aligned matrices when comparing with a wildtype model [45].

Yin Yang 1 (YY1), involved in transcriptional control, chromatin remodeling, and DNA damage repair, might also be targeted by miR-29c-5p. Although the role of YY1 in cancer is controversial, with both oncogenic and tumor suppressor roles [46], YY1 acts as a strong negative regulator of periostin expression [47]. The glycoprotein periostin regulates collagen fibril morphology and potentially LOX-mediated fibril crosslinking. Periostin is overexpressed in raised mammographic density and in most breast cancers, where it enhances angiogenesis and tumor progression, and recruits Wnt ligands to maintain cancer stem cell maintenance [48]. YY1 was however overexpressed in two 3D mammary epithelial cell models that mimic high MD [38].

Mostly based on research that focused on other miR-29 family members having an identical miRNA seed region (the miRNA region that is essential for the binding of the miRNA to the mRNA), one of the most interesting potential targets of hsa-miR-29c-5p in relation to high MD development is CD36. This transmembrane receptor expressed on the surface of breast stromal cells modulates multiple MD-related processes including adipocyte differentiation, fibroblast activation, matrix accumulation, cell–ECM interactions, and immune signaling [2,9,49,50,51]. CD36 downregulation in vitro causes accumulation of various ECM proteins including collagen type I and fibronectin. In vivo, CD36 knockout mice show less fat accumulation and more ECM accumulation, two prominent phenotypes observed in desmoplasia and high MD [8]. Consequently, low stromal CD36 expression has an established relation to high MD [8].

Future functional studies are needed to confirm suggested targets, to explore the associations with MD phenotype and breast cancer, to determine whether miRNA dysregulation is a cause or a consequence of high MD, or whether miRNA dysregulation and high MD are a consequence of another common factor. This study, for example, also showed elevated miR-29c-5p expression in nulliparous women. It has been demonstrated by other research that parity can affect miRNA expression [52], however there is no in depth reason as to why and how this takes place. Estrogen changes due to pregnancy likely influence the miRNA expression due to genomic and non-genomic mechanisms of action [53], but whether these miRNA changes are temporary has not been investigated.

From a research point of view, there are some limitations to this study. Firstly, only 754 miRNAs out of more than 2600 established human mature miRNAs were investigated [54], and only 118 of these 754 miRNAs were reliably expressed in NAF. Other potentially relevant miRNAs may thus not have been discovered here. In addition, these miRNAs were measured in NAF only, so defining their origin cells was not possible and future studies are necessary to discover the source of the miRNAs. Furthermore, statistical analysis of the discovery cohort was hampered by insufficient power, leading to adjusted strategies that may have affected the results. Future studies therefore require careful external and functional validation of the candidate miRNAs. Lastly, ethnicity and menopausal influences were not considered in this study [55].

## 5. Conclusions

We identified and independently validated increased miR-29c-5p expression in extremely high MD NAF. Comparison of the two extreme density categories was hypothesized to result in the biggest difference in miRNA expression. The next step will be to functionally investigate its targets, known or expected, for their involvement in high MD and subsequent breast cancer development. Our results provide new insights into how high MD might arise and why it is associated with an increased breast cancer risk. An ongoing Dutch trial (NTR6162/NL6031) is now examining miR-29c-5p in NAF from breast cancer patients, in relation to MD. Ultimately, this might enable the development of a breast cancer risk classifier for women with high MD, and of targeted therapies to modify this risk.

## Figures and Tables

**Figure 1 cancers-14-03805-f001:**
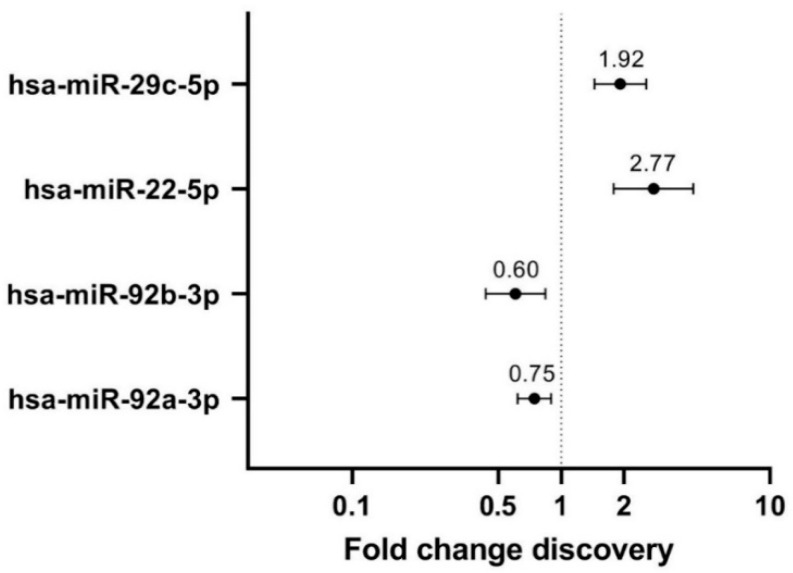
Linear regression-based fold changes in extremely high compared to very low mammographic density derived nipple aspirate fluid, of the four differentially expressed miRNAs in the discovery cohort. miRNAs with *p*-values < 0.2 were considered of interest for subsequent validation. miR-92b-3p (*p* = 0.139) and miR-92a-3p (*p* = 0.125) were negative predictors for extremely high MD (downregulated versus very low MD), whereas miR-22-5p (*p* = 0.027) and miR-29c-5p (*p* = 0.031) were positive predictors for extremely high MD (upregulated versus very low MD).

**Figure 2 cancers-14-03805-f002:**
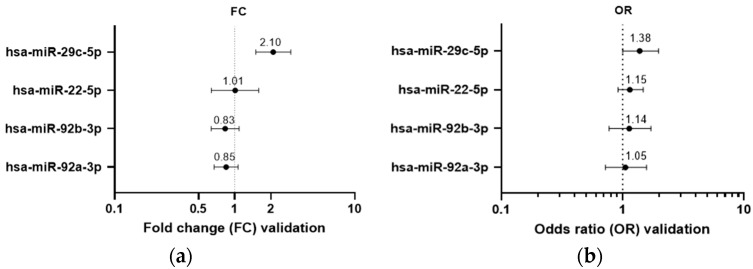
Fold change and forest plot representing the results of linear and logistic regression analysis with the four candidate differentially expressed human mature miRNAs in the validation cohort. (**a**) Fold miRNA expression change (FC) in extremely high versus very low MD, based on linear regression analysis of the four miRNAs. MiR-29c-5p was upregulated in high MD-derived nipple aspirate fluid (*p* = 0.029). (**b**) Odds ratios (OR) from the logistic regression analysis including each of the four human miRNAs individually. One miRNA independently predicted extremely high MD in NAF: miR-29c-5p (OR = 1.38 (95% CI 1.00–1.98); *p* = 0.048).

**Figure 3 cancers-14-03805-f003:**
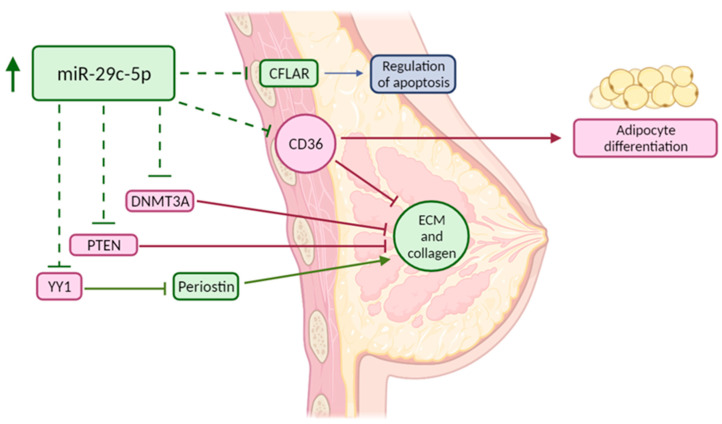
Graphical overview of miRNA–mRNA target interactions possibly related to the development of high mammographic density based on this study. Green targets and pathways are upregulated whereas red targets and pathways are downregulated, blue is an effect of the target. Created with BioRender.com. CFLAR = CASP8 and FADD-like apoptosis regulator; DNMT3A = DNA methyltransferase 3A; ECM and collagen = ECM organization and collagen deposition; PTEN = phosphatase and tensin homolog; YY1 = Yin Yang 1.

**Table 1 cancers-14-03805-t001:** Baseline characteristics of extremely high (“high”) and very low (“low”) mammographic density cohorts. (**a**) Discovery cohort and (**b**) validation cohort. Cohort size per baseline characteristic can differ due to missing values. Bold *p-*values indicate significant difference.

**(a) Discovery Cohort (N = 41)**				
		**High Density**	**Low Density**	***p* Value**
Age	Median (range)	N = 21 55 (51–72)	N = 20 54.5 (50–60)	0.29
BMI	Median (range)	N = 21 21.7 (18.4–28.9)	N = 17 27.6 (21.6–38.8)	**<0.0001**
Age at first live birth	Median (range)	N = 15 30 (21–34)	N = 18 29 (18–38)	0.81
Age at menarche	Median (range)	N = 21 13 (11–16)	N = 20 13 (9–18)	0.76
Parity	Nulliparous (*n* = 8)	6 (29%)	2 (10%)	0.24
	Parous (*n* = 33)	15 (71%)	18 (90%)	
First degree BC	Yes (*n* = 8)	4 (31%)	4 (22%)	0.69
	No (*n* = 23)	9 (69%)	14 (78%)	
NAF color	Clear white/yellow (*n* = 11)	3 (14%)	8 (40%)	**0.028**
	Turbid white/yellow (*n* = 3)	1 (5%)	2 (10%)	
	Bloody/orange/pink (*n* = 13)	11 (52%)	2 (10%)	
	Green/brown (*n* = 14)	6 (29%)	8 (40%)	
**(b) Validation Cohort (N = 170)**				
		**High Density**	**Low Density**	***p* Value**
Age	Median (range)	N = 89 56 (50–74)	N = 81 55 (50–60)	0.05
BMI	Median (range)	N = 84 21.8 (17.0–34.5)	N = 73 29.0 (23.1–49.6)	**<0.0001**
Age at first live birth	Median (range)	N = 67 28 (19–42)	N = 72 27 (20–42)	0.23
Age at menarche	Median (range)	N = 82 14 (10–17)	N = 80 13 (9–16)	**0.001**
Parity	Nulliparous (*n* = 25)	17 (20%)	8 (10%)	0.07
	Parous (*n* = 139)	67 (80%)	72 (90%)	
First degree BC	Yes (*n* = 26)	15 (25%)	11 (15%)	0.12
	No (*n* = 108)	44 (75%)	64 (85%)	
NAF color	Clear white/yellow (*n* = 79)	41 (47%)	38 (47%)	0.237
	Turbid white/yellow (*n* = 45)	22 (25%)	23 (28%)	
	Bloody/orange/pink (*n* = 31)	18 (20%)	13 (16%)	
	Green/brown (*n* = 14)	7 (8%)	7 (9%)	

**Table 2 cancers-14-03805-t002:** Relevant targets of breast density associated hsa-miR-29c-5p. Targets are confirmed by strong (luciferase reporter assays, Western blot, or qPCR) or weaker experimental evidence (next generation sequencing, microarray, or others, annotated with *). Protein class, Gene Ontology (GO) biological process (BP) and Reactome pathways were explored via PANTHER 16.0 pathway analysis (http://www.pantherdb.org/ (accessed on 23 February 2022)) and summarized.

Hsa-miR-29c-5p Targets	Protein Class	Relevant GO BP and Reactome Pathways
CPEB4	mRNA polyadenylation factor	regulation of translation, translational elongation, ionotropic glutamate receptor signaling pathway, response to ischemia
TMEM98	Transmembrane protein	protein localization to nucleus, protein processing, negative regulator of FRAT2 mediated Wnt/ß-catenin signaling
CD36 *	Membrane trafficking regulatory protein	positive regulation of NF-kappaB TF activity, Toll-like receptor cascades, regulation of ERK1/2 cascade, regulation of gene expression, regulation of cell death, regulation of cell-matrix adhesion, phagocytosis, immune response, transcriptional regulation of white adipocyte differentiation, triglyceride transport, fatty acid/lipid metabolic process, lipid storage
CFLAR *	Protease	positive regulation of ERK1 and ERK2 cascade, positive regulation of I-kappaB kinase/NF-kappaB signaling, apoptotic signaling pathway, regulation of necroptotic process, negative regulation of ROS biosynthetic process, negative regulation of cellular response to TGF-β stimulus, wound healing, cellular response to estradiol, testosterone, hypoxia and EGF stimulus, proteolysis, regulation of ECM organization
DNMT3A *	DNA methyltransferase	epigenetic regulation of gene expression, chromatin organization, metabolism of proteins, SUMOylation, mitotic cell cycle, response to estradiol, positive regulation of cell death, cellular response to hypoxia/ toxic substance
YY1 *	Transcription factor	(regulation of) DNA repair, estrogen-dependent gene expression, nucleotide excision repair, RNA localization, regulation of transcription, regulation of cell cycle
PTEN *	Protein phosphatase	negative regulation of PI3-kinase and AKT signaling, PDGFR signaling pathway, p53 pathway, regulation of apoptotic signaling pathway, canonical Wnt signaling pathway, regulation of ERK1 and ERK2 cascade, protein dephosphorylation, angiogenesis, regulation of cell population proliferation, response to glucose, regulation of gene expression, negative regulation of EMT, negative regulation of cell migration, response to estradiol/hypoxia/insulin-like growth factor stimulus, negative regulation of G1/S phase transition

AKT = PKB = protein kinase B; ECM = extracellular matrix, EGF = epidermal growth factor; EMT = epithelial to mesenchymal transition; ERK = extracellular signal-regulated protein kinase; NF = nuclear factor; PDGF = platelet derived growth factor; PI3 = phosphoinositide 3; ROS = reactive oxygen species; TF = transcription factor; TGF = transforming growth factor.

## Data Availability

Data supporting the results can be found in the Appendix A in this article.

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
