# Peer review of "Elevated miR-29c-5p Expression in Nipple Aspirate Fluid Is Associated with Extremely High Mammographic Breast Density"

_cancers, 2022, doi:10.3390/cancers14153805_

Round 1

Reviewer 1 Report

The authors present an interesting area of investigation into miRNA profiling that may be related to breast density, an established risk of breast cancer. Very little is known or understood about the genetic and biochemical processes that predispose some women towards dense breast tissue and consequently the paper represents a novel study to explore this important breast cancer variable. The results represent preliminary examination of a well-defined cohort, with well rationalised experiment design and statistical analysis of the miRNA data. The findings that one particular miRNA is significantly differentially expressed and is thought to be responsible for expression of mRNA that translates to proteins involved in ECM organisation and adipocyte differentiation, is highly relevant. The authors provide a balanced interpretation of the results, identifying limitations of the current study that will be addressed in the future.

Recommendation

Line 262 – The technical validation of 14 miRNAs was quoted as “0.68-0.93” but should be “0.37 – 0.93” (Supplementary Figure 3). In the light of the wide range, the Pearson coefficients, between the discovery and validation experiments, for the four candidate miRNAs, should also be included in the paper.

Minor corrections

Line 9 – part of the address for some of the authors seems to be missing

Line 113 – “RLT” - requires a definition

Line 163 – “sporadic missings” – is this the correct term? Need a definition. Does this mean the volunteer didn’t provide the information, the information was unknown to the volunteer or the information was not relevant to volunteer?

Line 167 – “were taken along” in “should be “were included in”

Line 280 – “candidate” should be “candidates”

Line 376 – “ECM proteins among which collagen type I and fibronectin” better would be “ECM proteins including collagen type I and fibronectin”

Supplementary Figure 1: “Logistic regression analysis with the four candidate differentially expressed human mature miRNAs in the discovery cohort.” should be, ”Logistic regression analysis with the four candidates differentially expressed human mature miRNAs in the discovery cohort.”

Supplementary Table 3 “category (extremly high or very low)” should be “category (extremely high or very low)” Definition for UD required

Reviewer 2 Report

In the submitted manuscript Vissers et al. showed that microRNA miR-29c-5p expression measured in nipple aspirate fluid (NAF) is associated with extremely high mammographic breast density, which is a known risk factor for breast cancer.

This manuscript is quite well written, experiments were properly designed and conducted, while conclusions were corroborated by obtained results.

There are several, mostly minor, drawbacks which have to be improved or corrected.

1) Since microRNA is not a synonym for 'miR-' part of microRNA's name, in title there should be complete name of that mature microRNA: miR-29c-5p

2) The text is little bit misleading because it is not obvious from the beginning that NAFs were obtained from healthy women. In fact, this was mentioned only once in line 72. Therefore, this must be stressed out in both 'Abstract' and 'Materials and Methods'.

3) Lines 92-93: It would be more intuitive if IQR was presented with 25th and 75th percentiles.

4) For ALL used web-based tools and databases provide their valid web address in the main text. Also cite references for ALL used web-based tools and databases (e.g., for TargetScan 8.0 and miRWalk are missing). Precisely state if any specific R-package was used (and its version number and cite its reference if published in a scientific journal).

5) Lines 163-164: Provide reference for method you used for presenting undetected expressions, since IMHO your methods is not commonly used (https://doi.org/10.1093/bioinformatics/btu239). Also, put more clear phrase "maximum CT value obtained for particular miRNA + 1".

6) Line 185: Explain what "DCT" abbreviation means.

7) Since delta Ct values were used for statistical analyses, explain in line 199 how "fold change" was calculated, I suppose by this method https://doi.org/10.1006/meth.2001.1262

8) Whenever you present odds ratio in the text or figure caption, ALWAYS provide its 95% CI, and not just p-value. This is especially important for Figure 2b, since for the only stat. significant miR-29c-5p it seems that it reaches 1.00! Also, present ORs more intuitively like 'OR (95%CI), p-value'.

9) In Table 1 and Supp. Table 4, stating for how many samples the data is missing in cells with variable's name is awkward.

10) Use "N=" for denoting the sample size for the complete cohort, while "n=" for sub-cohorts.

11) Reference 19 is inappropriate, it is just a random web page from on-line help. If there is no adequate journal paper, Thermo Fisher Cloud's manufacturer and web URL in the main text is sufficient.

12) Authors stated that there is an ongoing trial examining miR-29c-5p in NAF from breast cancer patients, but nevertheless, authors could inspect TCGA BRCA dataset to explore differential expression of that miRNA and its target genes and their prognostic significance using various user friendly web-based tools like GEPIA, cBioPortal, UCSC Xena Browser, UALCAN, OncoLnc, KM plotter, etc.
